# Surprisingly Popular Voting with Concentric Rank-Order Models

## Abstract

An important problem on social information sites is the recovery of ground truth from individual reports when the experts are in the minority. The wisdom of the crowd, i.e. the collective opinion of a group of individuals fails in such a scenario. However, the surprisingly popular (SP) algorithm [15] can recover the ground truth even when the experts are in the minority, by asking the individuals to report additional prediction reports–their beliefs about the reports of others. Several recent works have extended the surprisingly popular algorithm to an equivalent voting rule (SP-voting) to recover the ground truth ranking over a set of $m$ alternatives. However, we are yet to fully understand when SP-voting can recover the ground truth ranking, and if so, how many samples (votes and predictions) it needs. We answer this question by proposing two rank-order models and analyzing the sample complexity of SP-voting under these models. In particular, we propose concentric mixtures of Mallows and Plackett-Luce models with $G(\geq 2)$ groups. Our models generalize previously proposed concentric mixtures of Mallows models with 2 groups, and we highlight the importance of $G > 2$ groups by identifying three distinct groups (expert, intermediate, and non-expert) from existing datasets. Next, we provide conditions on the parameters of the underlying models so that SP-voting can recover ground-truth rankings with high probability, and also derive sample complexities under the same. We complement the theoretical results by evaluating SP-voting on simulated and real datasets.

## CCS Concepts

• **Information systems** → **Rank aggregation**; *Probabilistic retrieval models*.

## Keywords

Surprisingly Popular Voting, Mixture Models, Preference Aggregation

**ACM Reference Format:**

Anonymous Author(s). 2024. Surprisingly Popular Voting with Concentric Rank-Order Models. In . ACM, New York, NY, USA, 13 pages. https://doi.org/10.1145/nnnnnnn.nnnnnnn

## 1 Introduction

The recovery of ground truth from individual reports is one of the most vital aspects of social information sharing and online

discourse. The *wisdom of the crowds* phenomenon refers to the observation that the collective value of a group of noisy individual opinions can be used to recover the ground truth [7]. Such a collective value cancels out the biases of individual opinions when the number of participants is large and is often deployed to recover the ground truth on online polling and Q&A platforms (e.g. Reddit).

However, when the experts are in the minority, approaches that rely on the collective opinion of a group of individuals fail to recover the ground truth. The Surprisingly Popular (SP) algorithm [15] is a promising technique capable of recovering the ground truth even when experts are in the minority. In addition to asking individuals' opinion (aka *vote*), it asks them to predict how they believe the majority's answer is (aka *prediction*). The SP algorithm then picks the outcome which is *surprisingly popular* i.e. whose actual frequency in the votes is greater than its average predicted frequency. It provably recovers the ground truth as the number of individuals grows, even with a minority of experts.

This approach has been extended to voting rules, called *SP-voting*, in order to recover the ground truth rankings over a set of $m$ alternatives. The naive application of SP-algorithm to voting requires that individuals submit their prediction as a distribution over $m!$ possible permutation of alternatives, which implies that the amount of information elicited from each voter is exponential in $m$. Surprisingly, it was shown that SP-voting can effectively recover the ground truth in practice even when prediction is of size $m$ [10]. In particular, Hosseini et al. [9] has shown that eliciting the most likely top-alternative or ranking can provide a significant improvement compared to classical voting rules. Furthermore, SP-voting has been extended to partial ranks where the voters provide reports (votes and predictions) over subsets of size $k$ with $k \ll m$ [9].

While SP-voting has been shown to be effective in full or partial rankings, we are yet to fully understand when SP-voting can recover the ground truth ranking, and if so, how many samples (votes and predictions) it needs. To the best of our knowledge, this question is unexplored even for the basic SP algorithm. The main difficulty of analyzing such algorithms is that they are *non-parametric* i.e. they don't make any assumptions about the underlying distribution of votes and predictions, and it's not immediately clear what type of parametric models would be a good fit for real-world datasets and are also amenable to analysis under the surprisingly popular framework. For the setting of partial rankings, Hosseini et al. [9] performed a preliminary analysis of SP-voting under a mixture of Mallows model with two groups. However, we observe that the real datasets need more than two groups and more general rank-order models. Thus, we ask the following questions:

> What general rank-order models can explain ranking datasets (both votes and predictions) with a ground truth ranking? Furthermore, can we analyze SP-voting under such rank-order models, and determine its sample complexity, and conditions for identifying the ground truth ranking?

## 1.1 Our Contributions

We propose various rank-order models with a ground truth ranking, and analyse the SP-voting rule under these models. In particular, our contributions are the following.

- We propose two rank-order models, the Concentric Mixture of Mallows and the Concentric Mixture of Plackett-Luce, and generalize them to accommodate populations of $G \geq 2$ groups.
- We derive the conditions required for the identification of ground truth ranking under the SP-voting and the proposed concentric rank-order models. The derived conditions highlight a tension between the fraction of different groups and the "expertise" (i.e. noise levels) of different groups.
- To evaluate practical viability, we fit these models to real-world datasets for populations with $G = 2$ and $G = 3$ groups. When $G = 3$, besides the expert and non-expert groups, we identify an intermediate group of voters of large fraction that explains the observed datasets better than prior approaches with two groups.
- Furthermore, we generate synthetic data based on these models and provide empirical results on the sample complexity of SP-Voting, comparing it against the Copeland rule. Finally, experiments on real-world datasets show that SP-voting performs significantly better than the Copeland voting rule even when the dataset size is small.

## 1.2 Related Work

The challenge of ground truth recovery using the wisdom of the crowd has been extensively explored in social choice theory [6, 7, 17]. Several vote aggregation rules [1, 5, 6, 19] have been proposed based on this concept to aggregate voters' preferences and recover the underlying ground truth. However, this approach falters when the majority of participants are misinformed [16], biased [3], or when expert opinions are underrepresented within the population [15]. To address this limitation, Prelec et al. [15] introduced the Surprisingly Popular (SP) algorithm, which requires voters to provide two types of information: their individual vote and their prediction of the consensus vote. However, Prelec et al. [15]'s SP algorithm becomes impractical when the objective is to recover true ordinal ranking, since it necessitates information across all $m!$ possible vote configurations. Hosseini et al. [10] extended the surprising popular algorithm to recover full rankings while reducing its complexity to $\binom{m}{2}$ votes, making it more practical for smaller values of $m$. Further extending this line of work, Hosseini et al. [9] proposed algorithms that generalize SP-Voting to handle any number of alternatives, while also introducing mechanisms for partial preference elicitation to improve the efficiency of ground truth recovery. However, it is still unclear under what conditions SP-Voting is effective for a large number of alternatives when eliciting rankings. Specifically, the structure of the voting population and whether their voting behavior can be mathematically modeled need to be studied in detail.

The modeling of ranked data can be approached from two perspectives: modeling the population of voters and modeling the ranking process itself [13]. To date, the SP-Voting framework has been examined primarily by classifying voters into two distinct groups.

Our work extends this analysis by generalizing it to account for any number of groups, denoted as $G$. In terms of modeling the ranking process, several probabilistic models have been developed to represent voter preference generation. These include Order Statistic models, such as the Thurstonian model [18]; Pairwise Comparison models, like the Bradley-Terry model [2]; Multistage models, such as the Plackett-Luce model [11, 14]; and Distance-based models, like the Mallows' model [12], among others. Marden [13] provides a more comprehensive review of these models.

The SP-Voting framework was recently studied by Hosseini et al. [9] under the assumption that voters' preferences are drawn from an underlying probability distribution known as the Concentric Mixture of Mallows model, a variant of Mallows' model. In this work, we extend the SP-Voting framework by investigating two different vote distribution assumptions: the distance-based Mallows' model and the multistage Plackett-Luce model. Specifically, we build on prior work by extending the Mallows' model to account for $G$ groups, allowing for a more general analysis of voter populations. Additionally, we propose a novel Concentric Plackett-Luce Mixture model, a variant of the multistage Plackett-Luce model, which similarly incorporates $G$ groups.

## 2 Model

Here we formally introduce the setting and the necessary notations. We will first introduce surprisingly popular voting considering reports over full rankings, and then cover the setting with partial rankings. Let $A = \{a_1, a_2, ..., a_m\}$ be the set of $m$ possible alternatives. The set $\mathcal{L}(A)$ represents all possible complete rankings over the alternatives. Let $\sigma \in \mathcal{L}(A)$ represent a complete ranking of the $m$ possible alternatives. We assume that there is a true ranking by $\sigma^\star \in \mathcal{L}(A)$; which is drawn from a prior $P(\cdot)$ over $\mathcal{L}(A)$. Voter $i$ observes a ranking $\sigma_i$ that is assumed to be a noisy version of the ground truth ranking $\sigma^\star$. We will write $\Pr_s(\sigma_i \mid \sigma^\star)$ to denote the probability that the voter $i$ observes her ranking $\sigma_i$ given the ground truth ranking $\sigma^\star$.

Given voter $i$'s ranking $\sigma_i$ and the prior $P(\cdot)$, voter $i$ can compute the posterior distribution over the ground truth using the Bayes rule.

$$\Pr_g(\sigma^\star \mid \sigma_i) = \frac{\Pr_s(\sigma_i \mid \sigma^\star) \cdot P(\sigma^\star)}{\sum_{\sigma' \in \mathcal{L}(A)} \Pr_s(\sigma_i \mid \sigma') \cdot P(\sigma')} \quad (1)$$

Using the posterior over the ground truth, voter $i$ can also compute a distribution over the rankings observed by another voter.

$$\Pr_o(\sigma_j \mid \sigma_i) = \sum_{\sigma' \in \mathcal{L}(A)} \Pr_s(\sigma_j \mid \sigma') \cdot \Pr_g(\sigma' \mid \sigma_i) \quad (2)$$

The surprisingly popular algorithm asks voters to report their votes, and posterior over others' votes. For each ranking $\sigma'$, it then computes the frequency $f(\sigma') = \frac{1}{n} \sum_i \mathbf{1}[\sigma = \sigma']$, and posterior

$$h(\sigma \mid \sigma') = \frac{1}{|\{i : \sigma_i = \sigma'\}|} \sum_{i : \sigma_i = \sigma'} \Pr_o(\sigma \mid \sigma_i),$$

and finally picks the ranking with highest *prediction normalized votes*.[1]

$$\widehat{\sigma} \in \text{argmax}_\sigma \overline{V}(\sigma) = f(\sigma) \cdot \sum_{\sigma' \in \mathcal{L}(A)} \frac{h(\sigma' \mid \sigma)}{h(\sigma \mid \sigma')} \quad (3)$$

Hosseini et al. [10] observed that asking for full posterior over $m!$ rankings might be prohibitive and introduced *surprisingly popular voting* (SP-voting) that only asks voters about ranking according to the posterior.

We will also consider the setting when voters report partial rankings over subsets of size $k \ll m$. Let us fix a subset $T \subseteq A$ of size $k$. Then the probability of a partial ranking $\pi_i$ given the ground truth ranking $\sigma^\star$ is

$$\text{Pr}_s(\pi_i \mid \sigma^\star) = \sum_{\sigma : \sigma \triangleright \pi_i} \text{Pr}_s(\sigma \mid \sigma^\star) \quad$$

Here $\sigma \triangleright \pi_i$ means that the ranking $\sigma$ when restricted to the subset $T$ is $\pi_i$. We can also naturally extend definition 1 to define the posterior distribution given a partial ranking.

$$\text{Pr}_g(\sigma^\star \mid \pi_i) = \frac{\text{Pr}_s(\pi_i \mid \sigma^\star) \cdot P(\sigma^\star)}{\sum_{\sigma' \in \mathcal{L}(A)} \text{Pr}_s(\pi_i \mid \sigma') \cdot P(\sigma')} \quad (4)$$

Using the posterior over the ground truth, voter $i$ can also compute the distribution over partial rankings observed by another voter.

$$\text{Pr}_o(\pi_j \mid \pi_i) = \sum_{\sigma' \in \mathcal{L}(A)} \text{Pr}_s(\pi_j \mid \sigma') \cdot \text{Pr}_g(\sigma' \mid \pi_i) \quad (5)$$

Finally, we can compute the *prediction-normalized vote* (as defined in eq. (3) but over partial rankings) and pick the partial ranking $\widehat{\pi}$ over the subset $T$ with the maximum value. We are interested in extension of SP-voting to partial rankings as proposed by Hosseini et al. [9]. Namely, the *partial-SP* algorithm first applies SP-voting to a collection of subsets to recover ground truth partial rankings over these subsets, and then aggregates them using a voting rule [9].

In the next section, we describe in detail the exact distribution that $\text{Pr}_s$ takes to accurately model the voter behavior and reason about our choices.

## 3 Concentric Mixtures Models

Concentric Mixture Models are a class of probabilistic models used to represent how different groups within a population rank a set of alternatives, all relative to a single underlying ground truth ranking. These models capture variations in group behavior by incorporating parameters that reflect the degree and nature of each group's deviation from this central ranking. Our main goal in this section is to analyze the performance of SP-voting under different concentric mixture models, by first identifying the conditions required to identify the ground truth, and then providing upper bounds on the sample complexity of SP-voting. We begin with the *Concentric Mixture of Mallows Model* in Section 3.1 , followed by the *Concentric Mixture of Plackett-Luce Model* in Section 3.2, which is a new model proposed in this work.

---

[1]This is the direct application of SP algorithm [15] by considering $m!$ possible ground truths.

## 3.1 The Concentric Mixture of Mallows Model

The *Concentric Mixture of Mallows Model* (CMM) [4] uses a distance-based approach to quantify deviations from the central ranking. Specifically, group $g$'s ranking is modeled as a Mallows model with a group-specific dispersion parameter $\phi_g$, which controls the degree of expertise of the group. The following equation describes the ranking observed by a voter where the voting population has $G$ distinct groups:

$$\text{Pr}_s(\sigma \mid \sigma^\star) = \sum_{g=1}^{G} p_g \cdot \text{Pr}_s(\sigma \mid \sigma^\star, \phi_g) \quad (6)$$

Here $\sigma^\star$ is the underlying ground-truth ranking, and $\text{Pr}_s(\sigma \mid \sigma^\star, \phi_g)$ is the probability of a voter observing the ranking $\sigma$ given the ground-truth ranking $\sigma^\star$ and the dispersion parameter $\phi_g$ for group $g$. The parameter $p_g$ represents the probability of voter $i$ belonging to group $g$, where $\sum_{g=1}^{G} p_g = 1$. In the Concentric Mixture of Mallows model, the probability $\text{Pr}_s(\sigma \mid \sigma^\star, \phi_g)$ is defined as:

$$\text{Pr}_s(\sigma \mid \sigma^\star, \phi_g) = \frac{\phi_g^{d(\sigma, \sigma^\star)}}{Z(\phi_g, m)} \quad (7)$$

where $d(\sigma, \sigma^\star)$ is the Kendall-Tau distance between the observed ranking $\sigma$ and the central ranking $\sigma^\star$, and $Z(\phi_g, m)$ is the normalization constant that ensures that the probabilities sum to 1 across all possible rankings. We will assume that $\phi_1 \leq \phi_2 \leq \ldots \leq \phi_G$. Note that, a smaller value of the dispersion parameter implies that the group is more expert i.e. likely to observe a ranking closer to the ground truth ranking.

For the case of two groups (i.e. $G = 2$), Collas and Irurozki [4] analyzed the identifiability and sample complexity of the concentric mixture model under the Borda voting rule. Our first goal is to analyze the same model under the SP-Voting rule and an arbitrary number of groups. There are two main steps in the analysis of SP-Voting

(1) *Identification*: determine the condition needed to ensure

$$\overline{V}(\sigma^\star) \geq 2 \cdot \max_{\tau : d(\tau, \sigma^\star) \geq 1} \overline{V}(\tau),$$

so that maximizing prediction-normalized-vote returns the ground truth.

(2) *Sample Complexity*: when the identification condition holds, determine the number of samples necessary to ensure

$$\widehat{\overline{V}}(\sigma^\star) > \max_{\tau : d(\tau, \sigma^\star) \geq 1} \widehat{\overline{V}}(\tau),$$

so that maximizing the prediction-normalized votes from samples returns the ground truth.

For the setting of $G = 2$, Hosseini et al. [9] proved the following result regarding identifying the CMM model. [2]

LEMMA 3.1 (HOSSEINI ET AL. [9]). *Suppose $p_1 \leq 1/2$ and the following condition holds.*

$$\left(\frac{p_1}{1 - p_1}\right)^2 \geq 2 \cdot \frac{Z(\phi_2)^3}{Z(\phi_1)^2} \phi_1^{m(m-1)/2}$$

---

[2]Hosseini et al. [9] actually proved the results for the case of partial rankings, and $G = 2$. Here we state a simplified version for full rankings.

Then for any $\tau$ with $d(\tau, \sigma^\star) \geq 1$ we have $\overline{V}(\sigma^\star) \geq 2\overline{V}(\tau)$.

The above result says that if the non-experts are too noisy (i.e. $\phi_2 \gg \phi_1$) then the fraction of experts $p_1$ cannot be too small. Next we generalize the lemma for the case of arbitrary number of groups.

LEMMA 3.2. *Suppose the set $G$ can be partitioned into sets $G_1 = \{1, 2, \ldots, s\}$ and $G_2 = \{s + 1, \ldots, G\}$. Let $\alpha = \sum_{j \in G_1} p_j$ and the following condition holds.*

$$\frac{\alpha}{Z(\phi_s)} + \frac{1 - \alpha}{Z(\phi_G)} \geq 2 \left( \frac{\phi_s}{Z(\phi_1)} \alpha + \frac{\phi_G}{Z(\phi_{s+1})} (1 - \alpha) \right)$$

*Then we are guaranteed that $\overline{V}(\sigma^\star) \geq 2\overline{V}(\tau)$ for any $\tau$ such that $d(\tau, \sigma^\star) \geq 1$.*

The proof is provided in the appendix where we generalize lemma 3.1 and also simplify the conditions required for identification. One way to interpret the result is that when the experts are in the minority i.e. $\alpha \ll 1/2$ then we need $Z(\phi_{s+1}) \geq 2\phi_G Z(\phi_G)$ i.e. the dispersion parameter of the best non-expert should be sufficiently large. In the next subsection, we derive identifiability results under a different concentric mixture model, and then later provide sample complexity of SP-Voting under different rank-order models.

## 3.2 The Concentric Mixture of Plackett-Luce Model

In this subsection, we introduce the *Concentric Mixture of Plackett-Luce Model* (CMPL), which uses an element-specific probabilistic framework to rank alternatives based on their relative probabilities within each group. Specifically, group $g$'s ranking is models as a Plackett-Luce model with a group specific parameter vector $\theta_g \in \mathbb{R}_+^m$. As before, the following equation describes the ranking observed by a voter, where the voting population is divided into $G$ distinct groups:

$$\Pr_s(\sigma \mid \sigma^\star, \boldsymbol{\theta}) = \sum_{g=1}^{G} p_g \cdot \Pr_s(\sigma \mid \sigma^\star, \theta_g) \tag{8}$$

Here $\sigma^\star$ is the ground-truth ranking, and $\theta_g$ is the vector of strength parameters for group $g$. The parameter $p_g$ represents the probability that voter $i$ belongs to group $g$, where the mixture weights satisfy the constraint $\sum_{g=1}^{G} p_g = 1$. In the Concentric mixture of Plackett-Luce model, the probability $\Pr_s(\sigma \mid \sigma^\star, \theta_g)$ is defined as:

$$\Pr_s(\sigma \mid \sigma^\star, \theta_g) = \prod_{j=1}^{m} \frac{\theta_{g, \sigma^{\star-1}(\sigma(j))}}{\sum_{\ell=j}^{m} \theta_{g, \sigma^{\star-1}(\sigma(\ell))}} \tag{9}$$

Here, $\sigma(j)$ denotes the alternative assigned to the $j$-th position in the ranking $\sigma$, while $\sigma^{\star-1}(\sigma(j))$ denotes the position of the alternative $\sigma(j)$ in the ranking $\sigma^\star$. Equation (9) describes a Plackett-Luce model with ground truth $\sigma^\star$ and strength parameter vector $\theta_g$, as $\theta_{g, \sigma^{\star-1}(\sigma(j))}$ represents the strength parameter for that alternative within group $g$, and, the denominator, $\sum_{\ell=j}^{m} \theta_{g, \sigma^{\star-1}(\sigma(\ell))}$, ensures that the probability of selecting each alternative is normalized, considering only the alternatives that remain to be ranked.

### 3.2.1 Constraints on Strength Parameters.
Recall that in the concentric mixture of Mallows model the groups were ranked according to their dispersion parameters, i.e. $\phi_{g_1} \leq \phi_{g_2}$ implies that group $g_1$ is more expert compared to the group $g_2$. We now impose a similar condition on the parameters of the concentric mixture of Plackett-Luce model.

The strength parameters $\theta_g$ for each group are subject to two key constraints:

- **Within-group constraint**: For each group $g$, the sum of the strength parameters equals $1^3$, ensuring that the sum of the parameters is identical across the $G$ groups.

$$\sum_{j=1}^{m} \theta_{g,j} = 1 \quad \forall g \in \{1, \ldots, G\}.$$

Additionally, the entries in $\theta_g$ are non-increasing i.e. $\theta_{g,i} \geq \theta_{g,j}$ for $i \geq j$.

- **Between-group constraint**: The strength parameters for the higher-expertise group should stochastically dominate those of the lower-expertise groups. In particular, for any location $\ell$ the following condition must hold.

$$\sum_{j=1}^{\ell} \theta_{1,j} \geq \sum_{j=1}^{\ell} \theta_{2,j} \geq \cdots \geq \sum_{j=1}^{\ell} \theta_{G,j} \quad \forall \ell \in \{1, \ldots, m\}.$$

This hierarchical constraint ensures that the behavior of the groups is ordered in a way that reflects their relative strengths, with group 1 being closest to the ground-truth ranking, and subsequent groups deviating further from it.

We now turn to derive the identification condition to ensure that the ground truth ranking is the unique ranking to maximize the prediction-normalized vote. The next lemma gives a sufficient condition under the CMPL model and two groups.

LEMMA 3.3. *Suppose $p_1 \leq 1/2$ and the following condition holds.*

$$\left( \frac{p_1}{1 - p_1} \right)^2 \geq 2 \cdot \left( \prod_{j=1}^{m} \frac{\theta_{2,j}}{\sum_{i=j}^{m} \theta_{2,i}} \right) \left( \prod_{j=1}^{m} \frac{\theta_{1,j}}{\sum_{i=j}^{m} \theta_{1,i}} \right)^{-1} \left( \prod_{j=1}^{m} \frac{\theta_{1,m-j+1}}{\sum_{i=j}^{m} \theta_{1,m-i+1}} \right)$$

*Then for any ranking $\tau$ with $d(\tau, \sigma^\star) \geq 1$ we are guaranteed that $\overline{V}(\sigma^\star) \geq 2\overline{V}(\tau)$.*

In order to interpret the condition, let us choose a simple setting of strength parameters. Let $\theta_1 = (\gamma_1, 1, \ldots, 1)/(\gamma_1 + m - 1)$ and similarly $\theta_2 = (\gamma_2, 1, \ldots, 1)/(\gamma_2 + m - 1)$. Then it can be verified that condition of Lemma 3.3 simplifies to the following,

$$\left( \frac{p_1}{1 - p_1} \right)^2 \geq 2 \frac{\gamma_2}{\gamma_2 + m - 1} \frac{\gamma_1 + m - 1}{\gamma_1} \prod_{j=1}^{m-1} \frac{1}{\gamma_1 + m - j}$$

and for large enough $m$ we need $\frac{p_1}{1 - p_1} \gtrsim \sqrt{2\gamma_2/\gamma_1} \cdot m^{-(m-1)/2}$. This means that as $\gamma_2$ approaches $\gamma_1$ (i.e. non-experts become close to experts), we need a larger value of $p_1$ (i.e. fraction of experts) to succeed. The next lemma generalizes the identifiability condition to an arbitrary number of groups.

---

[3] The constant 1 can be arbitrary, but must be the same across the groups.

LEMMA 3.4. *Suppose the set $G$ can be partitioned into sets $G_1 = \{1, 2, \ldots, s\}$ and $G_2 = \{s+1, \ldots, G\}$. Let $\alpha = \sum_{j \in G_1} p_j$ and the following condition holds.*

$$\alpha \prod_{j=1}^{m} \frac{\theta_{s,j}}{\sum_{i=j}^{m} \theta_{s,i}} + (1-\alpha) \prod_{j=1}^{m} \frac{\theta_{G,j}}{\sum_{i=j}^{m} \theta_{G,i}}$$

$$\geq \frac{2\alpha \prod_{j=1}^{m} \frac{\theta_{1,j}}{\sum_{i=j}^{m} \theta_{1,i}} + 2(1-\alpha) \prod_{j=1}^{m} \frac{\theta_{s+1,j}}{\sum_{i=j}^{m} \theta_{s+1,i}}}{\alpha \cdot \prod_{j=1}^{m} \frac{\theta_{1,m-j+1}}{\sum_{i=j}^{m} \theta_{1,m-i+1}} + (1-\alpha) \cdot \prod_{j=1}^{m} \frac{\theta_{s+1,m-j+1}}{\sum_{i=j}^{m} \theta_{s+1,m-i+1}}}$$

*Then for any ranking $\tau$ with $d(\tau, \sigma^\star) \geq 1$ we are guaranteed that $\overline{V}(\sigma^\star) \geq 2\overline{V}(\tau)$.*

### 3.3 Sample Complexity Bounds

Once we have derived the identifiability conditions, the derivation of sample complexity is relatively straightforward. When the number of samples is large, the empirical prediction-normalized vote $\widehat{V}(\sigma)$ cocnentrates around $\overline{V}(\sigma)$ with high probability, and the condition $\overline{V}(\sigma^\star) \geq 2\overline{V}(\tau)$ guarantees that we can always ensure $\widehat{V}(\sigma^\star) \geq \widehat{V}(\tau)$ for any $\tau$ with $d(\tau, \sigma^\star) \geq 1$. Therefore, picking a ranking that maximizes the empirical prediction-normalized votes returns the ground truth ranking. The next lemma states the sample complexity for the CMM model.

LEMMA 3.5. *Under the same setting as Lemma 3.2, suppose the number of samples is $n \geq O\left(m!\sqrt{m\log(m/\delta)}\right)$. Then SP-voting recovers the ground truth ranking with probability at least $1 - \delta$.*

The proof is very similar to the proof of Corollary 1 from [9].

## 4 Experiments

In this section, we describe how we infer the parameters for both the CMM and the *CMPL* using a real-world dataset.

**Dataset.** We use a real-world dataset from a recent online experiment run on SP-voting by Hosseini et al. [9].[4] The dataset consists of real participants who provide both *Vote* and *Prediction* data across three distinct domains: *Geography, Movies,* and *Paintings.* The dataset contains rankings over five alternatives that are selected from a universe of 36 alternatives. The dataset contains reports from 432 participants over 12 questions from each domain. The alternatives are ranked based on the following domain-specific metrics:

- **Countries**: Ranked by population.
- **Movies**: Ranked by gross lifetime box-office earnings.
- **Paintings**: Ranked by auction prices.

In addition to their *Votes* over these alternatives, each participant provides their *Prediction* report based on the posterior belief about another participant's votes. The types of prediction reports are based on ranking and can be *Top* (most likely alternative), *Rank* (most likely ranking), *Top-t* (approval of top $t$ alternatives). We fit both variants of the *Concentric Mixture Models*— Mallows and Plackett-Luce— to the dataset to infer the parameters governing the group-specific ranking behaviors. The objective is to capture

how different population groups deviate from a shared underlying ground truth ranking.

**Inference Methodology**. To estimate the parameters of the models, we employ a *Bayesian inference* approach, which allows us to estimate the posterior distributions of the parameters given the observed rankings. In particular, we use *No-U-Turn Sampling (NUTS)* [8], an advanced variant of Hamiltonian Monte Carlo (HMC), to sample from the posterior distribution of the parameters. By utilizing this sampling technique, we can obtain accurate estimates of the model parameters, such as the proportion parameters ($p_k$), dispersion parameters ($\phi_g$) for the Mallows model, and strength parameters ($\theta_g$) for the Plackett-Luce model, for each of the population groups. The use of NUTS also enables us to quantify the uncertainty in the parameter estimates, providing credible intervals for the inferred parameters. This is particularly important when analyzing real-world ranking data, as it allows us to account for variability across different population groups and rankings. Next we discuss the parameter inference for the CMM and CMPL models.

### 4.1 Concentric Mixture of Mallows

We fit the CMM with $G = 2$ and 3 groups to the dataset described earlier in this section. Below we describe the parameter inference procedure for $G = 3$ groups, the more general case. The three groups are categorized as experts, intermediates, and non-experts. We infer several key parameters, including the proportion of each group ($p_k$), the dispersion parameters for experts' votes ($\phi_{E\text{-votes}}$) and predictions ($\phi_{E\text{-predictions}}$), the dispersion parameters for intermediates' votes ($\phi_{I\text{-votes}}$) and predictions ($\phi_{I\text{-predictions}}$), and the dispersion parameters for non-experts' votes ($\phi_{NE\text{-votes}}$) and predictions ($\phi_{NE\text{-predictions}}$).

We first compute the *Kendall-Tau distances* between each participant's vote and prediction rankings and the ground-truth ranking. These Kendall-Tau distances ($\tau_{\text{votes}}$ and $\tau_{\text{predictions}}$) serve as a measure of how much each participant's rankings deviate from the central ground-truth ordering. The model's priors for the dispersion parameters and the group proportions are specified as follows:

$$p \sim \text{Dirichlet}(2, 2, 4)$$
$$\phi_{E\text{-votes}} \sim N(0.1, 0.2), \quad \phi_{E\text{-predictions}} \sim N(0.4, 0.3)$$
$$\phi_{I\text{-votes}} \sim N(0.4, 0.2), \quad \phi_{I\text{-predictions}} \sim N(0.4, 0.3)$$
$$\phi_{NE\text{-votes}} \sim N(0.8, 0.2), \quad \phi_{NE\text{-predictions}} \sim N(0.8, 0.3)$$

These priors represent our assumptions about the behavior of the three groups, where votes of experts are expected to have the tightest alignment with the ground-truth ranking (small dispersion), intermediates show moderate dispersion, and non-experts have the highest dispersion. On the other hand, the predictions of experts, intermediates, and non-experts have a lot of overlap, representing each voter's opinion of the consensus ranking.

The likelihood function is structured to account for the possibility that each participant could belong to any of the three groups. This implies that the observed Kendall-Tau distances for votes and predictions are modeled as a mixture of normal distributions in the rank space, and we can set a maximum likelihood estimation problem to infer various parameters. In particular, we run the NUTS algorithm with four chains, each consisting of 8000 iterations, with 2000 iterations reserved for warm-up.

---

[4]The dataset can be found here - https://github.com/amrit19/Surprisingly-Popular-Voting-Partial

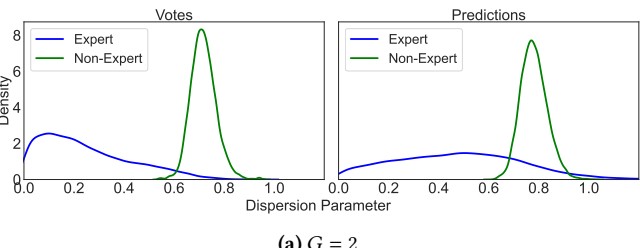
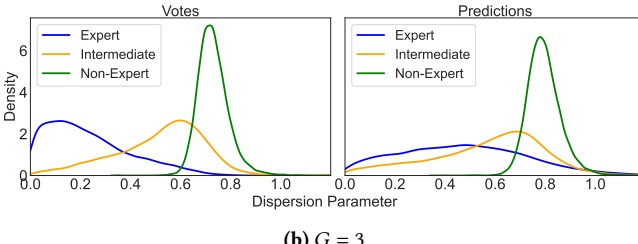

**(a)** $G = 2$                                                                 **(b)** $G = 3$

**Figure 1: Dispersion parameters for Votes and Predictions for $G = 2$ and $3$ of CMM model. We see that the CMM model with $G = 3$ identifies an intermediate group whose peak lies between experts and non-experts.**

Figure 1a and Figure 1b depict the distribution of dispersion parameters for Votes ($\phi_{\text{votes}}$) and Predictions ($\phi_{\text{predictions}}$) across different groups for $G = 2$ and $G = 3$. For votes, experts peak at a lower dispersion parameter in both $G = 2$ and $G = 3$, indicating more agreement, while non-experts peak at higher dispersion, showing greater spread in their voting. Experts show a widespread distribution for predictions since they reflect the majority belief, which deviates from the true belief, while non-experts are even farther away. The addition of the intermediate group in $G = 3$ adds valuable insights – their peak lies between experts and non-experts in votes, and their prediction distribution is similarly widespread as experts, reflecting the majority belief. This indicates that modeling voter behavior with more than two groups provides a more accurate and nuanced understanding of the data.

### 4.2 Concentric Mixture of Plackett-Luce

We fit the CMPL with $G = 2$ and $3$ groups. For $G = 3$, the groups are labeled as experts, intermediates, and non-experts. Similar to the CMM model, we infer the proportion of each group ($p_k$). Additionally, we infer the strength parameters for experts' votes ($\theta_{E\text{-votes}}$) and predictions ($\theta_{E\text{-predictions}}$), intermediates' votes ($\theta_{I\text{-votes}}$) and predictions ($\theta_{I\text{-predictions}}$), and non-experts' votes ($\theta_{NE\text{-votes}}$) and predictions ($\theta_{NE\text{-predictions}}$). We use the *Inference Method* described earlier in this section, utilizing the No-U-Turn Sampler (NUTS) to explore the parameter space and infer posterior distributions for the model parameters.

Before sampling, the rankings provided by participants (both votes and predictions) are converted into indices, which correspond to the options being ranked. The strength parameters, which reflect the relative probability of ranking an alternative higher than the others within a group, are inferred separately for experts, intermediates, and non-experts. The model's priors for the group proportions and the strength parameters are defined as follows:

$$p \sim \text{Dirichlet}(1, 2, 3),$$

$$\theta_{E\text{-votes}} \sim \text{Dirichlet}(\underbrace{3, 3, \ldots, 3}_{m}), \quad \theta_{E\text{-predictions}} \sim \text{Dirichlet}(\underbrace{1, 1, \ldots, 1}_{m}),$$

$$\theta_{I\text{-votes}} \sim \text{Dirichlet}(\underbrace{2, 2, \ldots, 2}_{m}), \quad \theta_{I\text{-predictions}} \sim \text{Dirichlet}(\underbrace{1, 1, \ldots, 1}_{m}),$$

$$\theta_{NE\text{-votes}} \sim \text{Dirichlet}(\underbrace{1, 1, \ldots, 1}_{m}), \quad \theta_{NE\text{-predictions}} \sim \text{Dirichlet}(\underbrace{1, 1, \ldots, 1}_{m})$$

These priors reflect the assumption that experts are expected to have higher strengths, indicating that they consistently rank the correct alternatives higher. Intermediates have moderate strengths, and non-experts are assumed to have the lowest strengths, indicating a less accurate ranking behavior.

In addition, we impose the model constraints described in Section 3.2.1, ensuring that the strength parameters for each group follow the expected relationships (e.g., ensuring that expert strengths are higher and decrease in a structured manner across groups). The likelihood function is structured to account for the mixture model, where participants may belong to one of the three groups. The observed rankings (in the form of indices) are used to compute the log-likelihood based on the Plackett-Luce model, where each group's strength parameters determine the probability of a particular ranking.

Similar to the CMM model, we run the NUTS algorithm with four chains, each consisting of 6000 iterations, with 2000 iterations reserved for warm-up. Figure 2a and Figure 2b show the distribution of strength parameters of Votes and Predictions for the first, third, and fifth positions in the ranking. We again observe the benefit of having $G = 3$ where the intermediate group peaks between the experts and non-experts (Figure 2b, Position 1). Additionally, the recovered strength parameters also demonstrate the stochastic dominance property. Looking at position 1 in both Figure 2a and Figure 2b, the strength parameter of the expert peaks at a higher value than the non-experts and intermediates. For positions 3 and 5 the peaks of the experts' strength parameter shifts left and gradually merges with non-experts, in order to ensure that $\sum_i \theta_{g,i} = 1$.

### 4.3 Predicting Complete Rankings from Partial Rankings using CMM and CMPL

We predict the complete ranking of 36 alternatives from partial rankings, for each population group (experts, intermediates, and non-experts) using the CMM and CMPL models. The dataset containing 36 alternatives is divided into 12 subsets, each containing 5 alternatives and we collect vote information over these subsets.

In both models, we use a hierarchical approach. We first fit each model to the subsets independently, learning the parameters for the alternatives within each subset. Since some alternatives appear in multiple subsets, this creates transitive relationships that help predict a global ranking across all 36 alternatives accurately. Once the parameters are inferred, we sample from the posterior

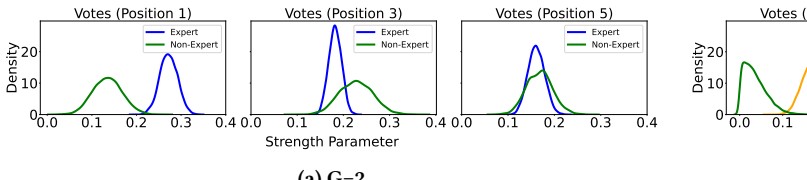

(a) G=2

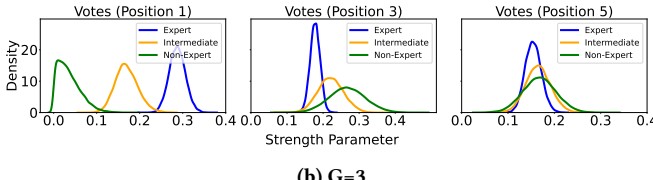

(b) G=3

Figure 2: Strength parameters for Votes at Positions 1, 3, and 5 for $G = 2$ and 3 of CMPL Model. We observe a stochastic dominance relationship. Initially, the strength parameter of the expert peaks at a large value, but gradually decreases at higher positions to ensure normalization.

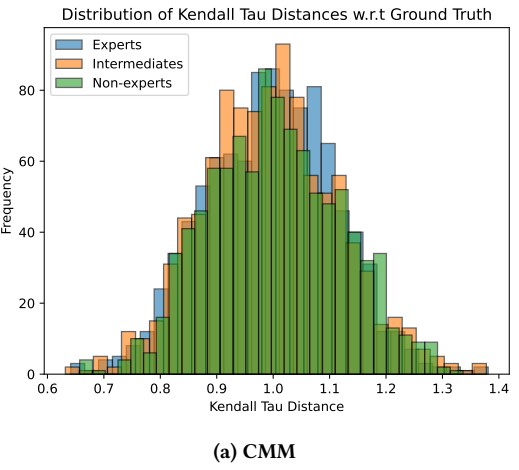

(a) CMM

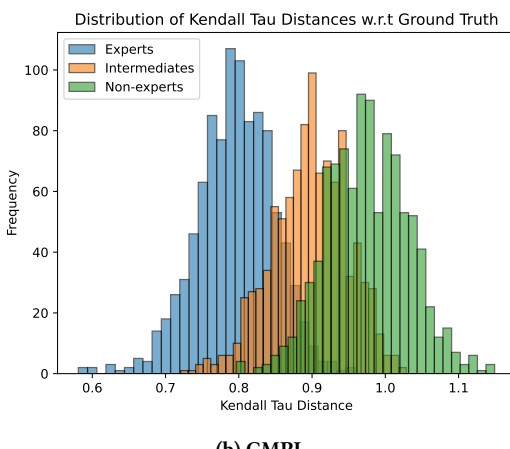

(b) CMPL

Figure 3: Distribution of complete rankings (predicted) for each population group. The data contains partial preferences over subsets. The CMPL model provides fine-grained inferences because it learns the weight of each position in the full ranking.

distributions and input these samples into the respective CMM or CMPL model to generate the full ranking.

**CMM.** For each subset, we infer the group-specific posterior distribution of dispersion parameters for each population group (experts, intermediates, and non-experts). Using these inferred parameters, we generate rankings by inputting the values into CMM model. This allows us to compute a distribution of Kendall Tau distances by comparing the predicted subset-level rankings to the ground truth for each group. We then sample from the posterior of these group-specific distributions- both the dispersion parameters and Kendall Tau distances- and use these samples in the CMM model to generate full rankings for all 36 alternatives. To quantify uncertainty in these predicted rankings, we apply bootstrapping, which provides a range of plausible full rankings derived from the posterior samples.

**CMPL.** For each subset, we infer the posterior distribution of group-specific strength parameters for each alternative, providing a probabilistic estimate of each alternative's rank. We use the CMPL model to iteratively select the alternative with the highest sampled strength parameter at each position, repeating the process for the remaining positions to generate a complete ranking. To quantify uncertainty, we apply bootstrapping, generating a full distribution of plausible complete rankings.

Figure 3a and Figure 3b show the distribution of Kendall Tau distance for each group (experts, intermediates, and non-experts) when the complete rankings are inferred from CMM and CMPL respectively. For the CMPL model, Figure 3b, the distributions reflect that experts are closest to the ground truth, followed by intermediates, and then non-experts. This distinction is less pronounced in CMM model, Figure 3a. The CMPL model provides more fine-grained inferences because it learns the distribution over each position in the full ranking through the posterior estimates, allowing for more precise predictions of the rank order of alternatives. In contrast, the CMM model is less fine-grained, as it estimates how close the ranking is to the ground truth based on a single dispersion parameter, per population group, that represents the overall distance but lacks detailed information about specific positions within the ranking.

## 5 Sample Complexity Results

In this section, we analyze the impact of sample size on ground truth recovery by generating synthetic data using the CMM and CMPL models with $G = 3$. We generate 500 samples with the proportion of experts in the population being 1%. Figure 4 present a comparison of how sample size affects the performance of two aggregation methods: Copeland Rule [5] and SP-Voting. Figure 5 shows the same comparison on real data. Refer to Figure 6 in Appendix B for results with $G = 2$.

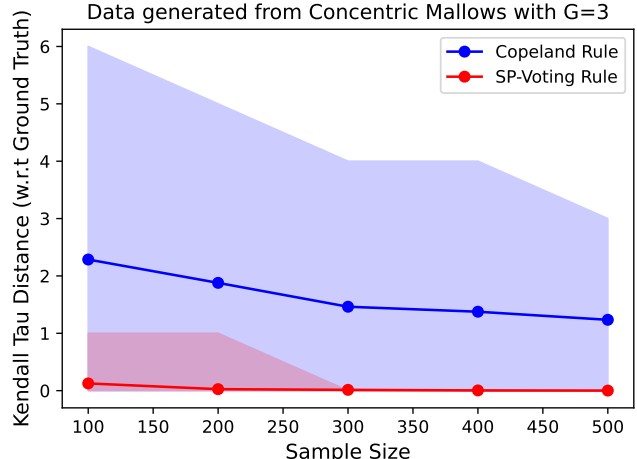

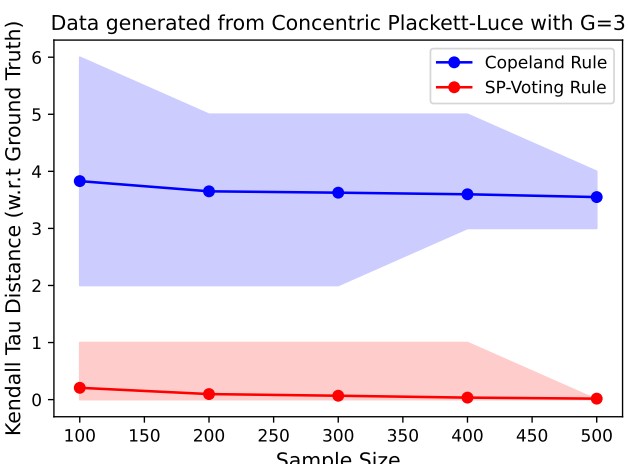

**Figure 4: Comparison of sample complexity for data generated from `CMM` and `CMPL` models with G=3, and aggregated using Copeland and SP-Voting rule.**

From Figure 4, it is evident that SP-Voting outperforms the Copeland Rule in terms of accurately recovering the ground-truth ranking as the sample size increases. For both CMM and CMPL models, the Kendall Tau distance between the estimated and ground truth rankings consistently decreases with increasing sample sizes. However, the SP-Voting method shows a sharper decline compared to the Copeland Rule, indicating its superior performance in reaching the ground truth. The confidence intervals (shaded areas) for SP-Voting are consistently narrower compared to those for Copeland, implying higher stability and lower variability of SP-Voting across different sampling scenarios. Figure 5 shows the analysis on a limited 48 samples of real data, where we can see a gradual decrease in the mean value and the confidence around it for SP-Voting as compared to Copeland, indicating that with more samples, ground-truth recovery can be achieved faster and with higher certainty using SP-Voting.

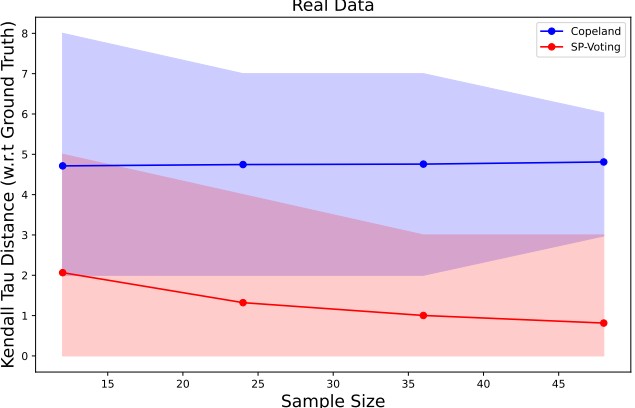

**Figure 5: Comparison of sample complexity on real data when votes are aggregated using Copeland and SP-Voting.**

Overall, the comparison between the two models— CMM and CMPL—shows similar trends, with SP-Voting consistently outperforming the Copeland Rule across both models. Increasing sample size notably helps both methods, but SP-Voting achieves ground truth recovery with fewer samples and more consistency. This indicates that the prediction information involved in SP-Voting helps correct the effect of non-expert votes and thus helps reach the ground truth faster. These findings reinforce the efficacy of SP-Voting over traditional aggregation rules like the Copeland Rule, in terms of both accuracy and reliability when aggregating rankings to recover the ground truth.

## 6 Discussion and Future Work

In this work, we have analyzed SP-voting under two concentric rank-order models (Mallows and Plackett-Luce) with an arbitrary number of groups. We observed that real-world datasets often have multiple groups of experts ($G \geq 3$) and SP-voting performs better in terms of sample complexity when compared to standard voting rules. There are many interesting directions for future work. First, Prelec et al. [15] have proposed the *self-predicting* property for the general SP algorithms. Although this condition is not sufficient to derive finite sample complexity bounds, it would be interesting to see how it compares with the conditions we derived for various concentric rank-order models. Second, we have seen that moving from $G = 2$ to $G = 3$ groups gives a significantly better fit (and explanation) with respect to the real data but the improvement is marginal for larger values of $G$. Then a natural question is can we choose the number of groups $G$ in a a data-dependent way? Finally, in terms of sample complexity, we have analyzed SP-voting for recovering ground truth ranking over $m$ alternatives, and the bound grows with $m!$. This can be reduced to $O(m^2)$ for the pairwise version of SP-voting considered in prior work [10] with additional assumptions. However, when the number of alternatives $m$ is large, we want the sample complexity to be independent of $m$. SP-voting with partial preferences [9] help in such contexts, and we leave a fine-grained analysis of the partial variants of SP (under various concentric rank-order models) as future work.

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

## A Missing Proofs

### A.1 Proof of Lemma 3.2

PROOF. We can proceed similar to the proof of Lemma 2 from [9] and establish the following upper and lower bounds on prediction normalized vote.

$$\frac{f(\sigma)}{\sum_{\tilde{\sigma}} \Pr_s(\sigma|\tilde{\sigma})} \le \overline{V}(\sigma) \le \frac{f(\sigma)}{\min_{\tilde{\sigma}} \Pr_s(\sigma|\tilde{\sigma})}$$

We can express the probability $\Pr_s(\sigma^{\star}|\tilde{\sigma})$ as follows

$$\Pr_s(\sigma|\tilde{\sigma}) = \sum_{j \in G_1} p_j \cdot \frac{\phi_j^{d(\tilde{\sigma},\sigma)}}{Z(\phi_j)} + \sum_{j \in G_2} p_j \cdot \frac{\phi_j^{d(\tilde{\sigma},\sigma)}}{Z(\phi_j)}$$

This gives us the following lower bound on $\overline{V}(\sigma^{\star})$.

$$\overline{V}(\sigma^{\star}) = \frac{\sum_{j \in G_1} \frac{p_j}{Z(\phi_j)} + \sum_{j \in G_2} \frac{p_j}{Z(\phi_j)}}{\sum_{\tilde{\sigma}} \left( \sum_{j \in G_1} p_j \cdot \frac{\phi_j^{d(\tilde{\sigma},\sigma^{\star})}}{Z(\phi_j)} + \sum_{j \in G_2} p_j \cdot \frac{\phi_j^{d(\tilde{\sigma},\sigma^{\star})}}{Z(\phi_j)} \right)}$$

$$\ge \frac{\frac{1}{Z(\phi_s)} \sum_{j \in G_1} p_j + \frac{1}{Z(\phi_G)} \sum_{j \in G_2} p_j}{\sum_{j \in G_1} p_j \sum_{\tilde{\sigma}} \frac{\phi_j^{d(\sigma^{\star},\tilde{\sigma})}}{Z(\phi_j)} + \sum_{j \in G_2} p_j \sum_{\tilde{\sigma}} \frac{\phi_j^{d(\sigma^{\star},\tilde{\sigma})}}{Z(\phi_j)}}$$

$$= \frac{\frac{\alpha}{Z(\phi_s)} + \frac{1-\alpha}{Z(\phi_G)}}{\sum_{j \in G_1} p_j + \sum_{j \in G_2} p_j}$$

$$= \frac{\alpha}{Z(\phi_s)} + \frac{1-\alpha}{Z(\phi_G)}$$

We can also obtain the following upper bound on $\overline{V}(\tau)$.

$$\overline{V}(\tau) \le \frac{\sum_{j \in G_1} p_j \cdot \frac{\phi_j}{Z(\phi_j)} + \sum_{j \in G_2} p_j \cdot \frac{\phi_j}{Z(\phi_j)}}{\sum_{\tilde{\sigma}} \left( \sum_{j \in G_1} p_j \cdot \frac{\phi_j^{d(\tilde{\sigma},\tau)}}{Z(\phi_j)} + \sum_{j \in G_2} p_j \cdot \frac{\phi_j^{d(\tilde{\sigma},\tau)}}{Z(\phi_j)} \right)}$$

$$\le \frac{\frac{\phi_s}{Z(\phi_1)} \sum_{j \in G_1} p_j + \frac{\phi_G}{Z(\phi_{s+1})} \sum_{j \in G_2} p_j}{\sum_{j \in G_1} p_j \sum_{\tilde{\sigma}} \frac{\phi_j^{d(\tilde{\sigma},\tau)}}{Z(\phi_j)} + \sum_{j \in G_2} p_j \cdot \sum_{\tilde{\sigma}} \frac{\phi_j^{d(\tilde{\sigma},\tau)}}{Z(\phi_j)}}$$

$$= \frac{\phi_s}{Z(\phi_1)} \alpha + \frac{\phi_G}{Z(\phi_{s+1})} (1 - \alpha)$$

Therefore, in order to ensure $\overline{V}(\sigma^{\star}) \ge 2\overline{V}(\tau)$ we need the following condition.

$$\frac{\alpha}{Z(\phi_s)} + \frac{1-\alpha}{Z(\phi_G)} \ge 2 \left( \frac{\phi_s}{Z(\phi_1)} \alpha + \frac{\phi_G}{Z(\phi_{s+1})} (1 - \alpha) \right)$$

□

### A.2 Proof of Lemma 3.3

PROOF. We can proceed similarly to the proof of Lemma 2 from Hosseini et al. [9] and establish the following upper and lower bounds on prediction normalized vote.

$$\frac{f(\sigma)}{\sum_{\tilde{\sigma}} \Pr_s(\sigma \mid \tilde{\sigma})} \le \overline{V}(\sigma) \le \frac{f(\sigma)}{\min_{\tilde{\sigma}} \Pr_s(\sigma \mid \tilde{\sigma})} \tag{10}$$

Suppose $\sigma^{\star}$ is the true ranking and consider any ranking $\tau$ with $d(\tau, \sigma^{\star}) \ge 1$. Without loss of generality, we can assume that $\sigma^{\star} = 1 \succ 2 \succ \ldots \succ m$. This also implies that $\theta_{g,1} \ge \theta_{g,2} \ge \ldots \ge \theta_{g,m}$ for any group $g$.

Under the assumption of Concentric mixture of Plackett-Luce model we have,

$$\Pr_s(\sigma^{\star} \mid \tilde{\sigma}) = p \cdot \Pr_s(\sigma^{\star} \mid \theta_1, \tilde{\sigma}) + (1 - p) \cdot \Pr_s(\sigma^{\star} \mid \theta_2, \tilde{\sigma})$$

$$= p \cdot \prod_{j=1}^{m} \frac{\theta_{1,\tilde{\sigma}^{-1}(\sigma^{\star}(j))}}{\sum_{i=j}^{m} \theta_{1,\tilde{\sigma}^{-1}(\sigma^{\star}(i))}} + (1 - p) \cdot \prod_{j=1}^{m} \frac{\theta_{2,\tilde{\sigma}^{-1}(\sigma^{\star}(j))}}{\sum_{i=j}^{m} \theta_{2,\tilde{\sigma}^{-1}(\sigma^{\star}(i))}}$$

When $\theta_1$ stochastically dominates $\theta_2$ we have $\Pr_s(\sigma^\star \mid \theta_1, \sigma^\star) \geq \Pr_s(\sigma^\star \mid \theta_2, \sigma^\star)$. Moreover, using the fact $p < (1-p)$ we obtain the following lower bound on $\overline{V}(\sigma^\star)$.

$$\overline{V}(\sigma^\star) \geq \frac{2p \cdot \prod_{j=1}^m \frac{\theta_{2,j}}{\sum_{i=j}^m \theta_{2,i}}}{(1-p) \cdot \sum_{\tilde{\sigma}} \prod_{j=1}^m \frac{\theta_{1,\tilde{\sigma}^{-1}(j)}}{\sum_{i=j}^m \theta_{1,\tilde{\sigma}^{-1}(i)}} + \prod_{j=1}^m \frac{\theta_{2,\tilde{\sigma}^{-1}(j)}}{\sum_{i=j}^m \theta_{2,\tilde{\sigma}^{-1}(i)}}} = \frac{p}{1-p} \cdot \prod_{j=1}^m \frac{\theta_{2,j}}{\sum_{i=j}^m \theta_{2,i}}$$

The last equality uses lemma A.1. We now provide an upper bound on $\overline{V}(\tau)$.

$$\Pr_s(\tau \mid \sigma^\star) = p \cdot \Pr_s(\tau \mid \theta_1, \sigma^\star) + (1-p) \cdot \Pr_s(\tau \mid \theta_2, \sigma^\star)$$
$$\leq p \cdot \Pr_s(\sigma^\star \mid \theta_1, \sigma^\star) + (1-p) \cdot \Pr_s(\sigma^\star \mid \theta_2, \sigma^\star)$$
$$\leq 2(1-p) \cdot \Pr_s(\sigma^\star \mid \theta_1, \sigma^\star)$$

The first inequality follows because the elements of $\theta_1$ and $\theta_2$ are arranged in non-decreasing order. The second inequality follows because $\theta_1$ stochastically dominates $\theta_2$. On the other hand,

$$\min_{\tilde{\sigma}} \Pr_s(\tau \mid \tilde{\sigma}) \geq p \left( \prod_{j=1}^m \frac{\theta_{1,m-j+1}}{\sum_{i=j}^m \theta_{1,m-i+1}} + \prod_{j=1}^m \frac{\theta_{2,m-j+1}}{\sum_{i=j}^m \theta_{2,m-i+1}} \right)$$
$$\geq 2p \cdot \prod_{j=1}^m \frac{\theta_{1,m-j+1}}{\sum_{i=j}^m \theta_{1,m-i+1}}$$

The last inequality follows since $\theta_1$ stochastically dominates $\theta_2$. Now we have the following upper bound on $\overline{V}(\tau)$.

$$\overline{V}(\tau) \leq \frac{1-p}{p} \cdot \frac{\prod_{j=1}^m \frac{\theta_{1,j}}{\sum_{i=j}^m \theta_{1,i}}}{\prod_{j=1}^m \frac{\theta_{1,m-j+1}}{\sum_{i=j}^m \theta_{1,m-i+1}}}$$

Therefore, as long as

$$\left( \frac{p}{1-p} \right)^2 \geq 2 \cdot \left( \prod_{j=1}^m \frac{\theta_{2,j}}{\sum_{i=j}^m \theta_{2,i}} \right) \left( \prod_{j=1}^m \frac{\theta_{1,j}}{\sum_{i=j}^m \theta_{1,i}} \right)^{-1} \left( \prod_{j=1}^m \frac{\theta_{1,m-j+1}}{\sum_{i=j}^m \theta_{1,m-i+1}} \right)$$

we are guaranteed that $\overline{V}(\sigma^\star) \geq 2\overline{V}(\tau)$. □

LEMMA A.1. *For any vector $u = (u_1, \ldots, u_m)$ we have,*

$$\sum_{\tilde{\sigma}} \prod_{j=1}^m \frac{u_{\tilde{\sigma}^{-1}(j)}}{\sum_{i=j}^m u_{\tilde{\sigma}^{-1}(i)}} = 1$$

PROOF. We prove this result by induction on $m$. For $m = 1$, there is only one permutation and the base case holds. Suppose, the claim is true for $m$. Then we have,

$$\sum_{\tilde{\sigma}} \prod_{j=1}^{m+1} \frac{u_{\tilde{\sigma}^{-1}(j)}}{\sum_{i=j}^{m+1} u_{\tilde{\sigma}^{-1}(i)}} = \sum_a \sum_{\tilde{\sigma}: \tilde{\sigma}[1]=a} \prod_{j=1}^{m+1} \frac{u_{\tilde{\sigma}^{-1}(j)}}{\sum_{i=j}^{m+1} u_{\tilde{\sigma}^{-1}(i)}}$$
$$= \sum_a \frac{u_a}{\sum_{j=1}^{m+1} u_j} \sum_{\tilde{\sigma}: \tilde{\sigma} \in \mathcal{S}} \prod_{j=1}^m \frac{u_{\tilde{\sigma}^{-1}(j)}}{\sum_{i=j}^{m+1} u_{\tilde{\sigma}^{-1}(i)}}$$
$$= \sum_a \frac{u_a}{\sum_{j=1}^{m+1} u_j} = 1$$

□

## A.3 Proof of Lemma 3.4

PROOF. We can proceed similarly to the proof of Lemma 2 from Hosseini et al. [9] and establish the following upper and lower bounds on prediction normalized vote.

$$\frac{f(\sigma)}{\sum_{\tilde{\sigma}} \Pr_s(\sigma \mid \tilde{\sigma})} \leq \overline{V}(\sigma) \leq \frac{f(\sigma)}{\min_{\tilde{\sigma}} \Pr_s(\sigma \mid \tilde{\sigma})} \tag{11}$$

Suppose $\sigma^\star$ is the true ranking and consider any ranking $\tau$ with $d(\tau, \sigma^\star) \geq 1$. Without loss of generality, we can assume that $\sigma^\star = 1 \succ 2 \succ \ldots \succ m$. This also implies that $\theta_{g,1} \geq \theta_{g,2} \geq \ldots \geq \theta_{g,m}$ for any group $g$.

Under the assumption of Concentric mixture of Plackett-Luce model we have,

$$\Pr_s(\sigma^\star \mid \tilde{\sigma}) = \sum_{\ell=1}^{G} p_\ell \cdot \Pr_s(\sigma^\star \mid \theta_\ell, \tilde{\sigma})$$

$$= \sum_{\ell=1}^{G} p_\ell \cdot \prod_{j=1}^{m} \frac{\theta_{\ell,\tilde{\sigma}^{-1}(\sigma^\star(j))}}{\sum_{i=j}^{m} \theta_{\ell,\tilde{\sigma}^{-1}(\sigma^\star(i))}}$$

$$= \sum_{\ell=1}^{G} p_\ell \cdot \prod_{j=1}^{m} \frac{\theta_{\ell,\tilde{\sigma}^{-1}(j)}}{\sum_{i=j}^{m} \theta_{\ell,\tilde{\sigma}^{-1}(i)}}$$

When $\theta$ stochastically dominates $\theta'$ we have $\Pr_s(\sigma^\star \mid \theta, \sigma^\star) \geq \Pr_s(\sigma^\star \mid \theta', \sigma^\star)$. This gives us the following lower bound on $\overline{V}(\sigma^\star)$.

$$\overline{V}(\sigma^\star) \geq \frac{\sum_{\ell \in G_1} p_\ell \cdot \prod_{j=1}^{m} \frac{\theta_{s,j}}{\sum_{i=j}^{m} \theta_{s,i}} + \sum_{\ell \in G_2} p_\ell \cdot \prod_{j=1}^{m} \frac{\theta_{G,j}}{\sum_{i=j}^{m} \theta_{G,i}}}{\sum_\ell p_\ell \cdot \sum_{\tilde{\sigma}} \prod_{j=1}^{m} \frac{\theta_{\ell,\tilde{\sigma}^{-1}(j)}}{\sum_{i=j}^{m} \theta_{\ell,\tilde{\sigma}^{-1}(i)}}}$$

$$= \alpha \prod_{j=1}^{m} \frac{\theta_{s,j}}{\sum_{i=j}^{m} \theta_{s,i}} + (1-\alpha) \prod_{j=1}^{m} \frac{\theta_{G,j}}{\sum_{i=j}^{m} \theta_{G,i}}$$

The last equality uses lemma A.1. We now provide an upper bound on $\overline{V}(\tau)$.

$$\Pr_s(\tau \mid \sigma^\star) = \sum_{\ell=1}^{G} p_\ell \cdot \Pr_s(\tau \mid \theta_\ell, \sigma^\star)$$

$$= \sum_{\ell=1}^{s} p_\ell \cdot \Pr_s(\tau \mid \theta_\ell, \sigma^\star) + \sum_{\ell=s+1}^{G} p_\ell \cdot \Pr_s(\tau \mid \theta_\ell, \sigma^\star)$$

Now using the stochastic dominance relation, we obtain the lower bound.

$$\min_{\tilde{\sigma}} \Pr_s(\tau \mid \tilde{\sigma}) \geq \sum_{\ell=1}^{s} p_\ell \prod_{j=1}^{m} \frac{\theta_{1,m-j+1}}{\sum_{i=j}^{m} \theta_{1,m-i+1}} + \sum_{\ell=s+1}^{G} p_\ell \prod_{j=1}^{m} \frac{\theta_{s+1,m-j+1}}{\sum_{i=j}^{m} \theta_{s+1,m-i+1}}$$

$$\geq \alpha \cdot \prod_{j=1}^{m} \frac{\theta_{1,m-j+1}}{\sum_{i=j}^{m} \theta_{1,m-i+1}} + (1-\alpha) \cdot \prod_{j=1}^{m} \frac{\theta_{s+1,m-j+1}}{\sum_{i=j}^{m} \theta_{s+1,m-i+1}}$$

Now we have the following upper bound on $\overline{V}(\tau)$.

$$\overline{V}(\tau) \leq \frac{\alpha \prod_{j=1}^{m} \frac{\theta_{1,j}}{\sum_{i=j}^{m} \theta_{1,i}} + (1-\alpha) \prod_{j=1}^{m} \frac{\theta_{s+1,j}}{\sum_{i=j}^{m} \theta_{s+1,i}}}{\alpha \cdot \prod_{j=1}^{m} \frac{\theta_{1,m-j+1}}{\sum_{i=j}^{m} \theta_{1,m-i+1}} + (1-\alpha) \cdot \prod_{j=1}^{m} \frac{\theta_{s+1,m-j+1}}{\sum_{i=j}^{m} \theta_{s+1,m-i+1}}}$$

Therefore, as long as

$$\alpha \prod_{j=1}^{m} \frac{\theta_{s,j}}{\sum_{i=j}^{m} \theta_{s,i}} + (1-\alpha) \prod_{j=1}^{m} \frac{\theta_{G,j}}{\sum_{i=j}^{m} \theta_{G,i}} \geq \frac{2\alpha \prod_{j=1}^{m} \frac{\theta_{1,j}}{\sum_{i=j}^{m} \theta_{1,i}} + 2(1-\alpha) \prod_{j=1}^{m} \frac{\theta_{s+1,j}}{\sum_{i=j}^{m} \theta_{s+1,i}}}{\alpha \cdot \prod_{j=1}^{m} \frac{\theta_{1,m-j+1}}{\sum_{i=j}^{m} \theta_{1,m-i+1}} + (1-\alpha) \cdot \prod_{j=1}^{m} \frac{\theta_{s+1,m-j+1}}{\sum_{i=j}^{m} \theta_{s+1,m-i+1}}}$$

we are guaranteed that $\overline{V}(\sigma^\star) \geq 2\overline{V}(\tau)$. □

# B  Missing Results

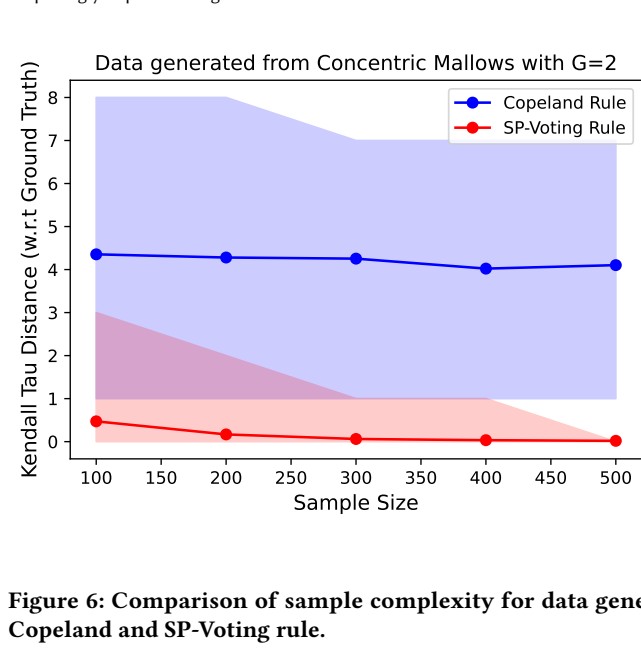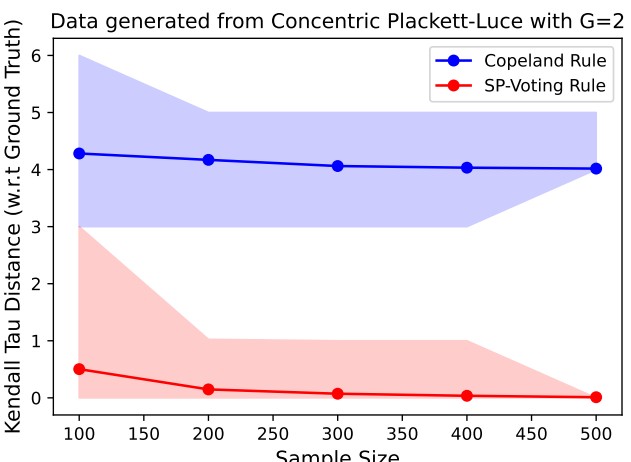

**Figure 6: Comparison of sample complexity for data generated from CMM and CMPL models with G=2, and aggregated using Copeland and SP-Voting rule.**

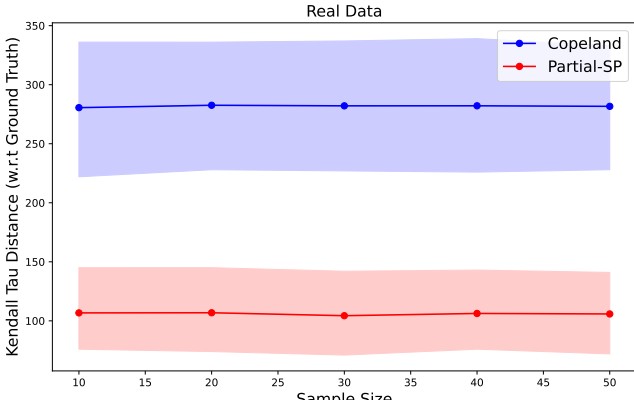

**Figure 7: Comparison of sample complexity on real data when votes are aggregated using Copeland and Partial-SP.**

