# OpenReview forum: "Surprisingly Popular Voting with Concentric Rank-Order Models"
_ACM.org/TheWebConf/2025/Conference — WWW 2025 Poster_

### Official Review · Reviewer_QFCV · 2024-11-25

**Novelty:** 4
**Technical Quality:** 4

**Review:**

Summary:

In this paper, the authors investigated the performance of SP-voting rule in two rank-order models---concentric mixtures of Mallows and Plackett-Luce models with $G > 2$ groups. The authors theoretically derived the conditions under which the SP-voting rule can identify the ground-truth ranking for each of these models. They further assessed the effectiveness of the SP-voting rule by fitting the models to real-world data, demonstrating its capability to accurately recover the ground-truth ranking.

Strength:

Overall, the paper is well-written and organized. The theoretical results appear sound and provide nice insights into the conditions under which the SP-voting rule would be effective. The authors also conducted extensive numerical experiments to validate their models.

Weakness & Questions:

Several theoretical and numerical results in the paper would benefit from additional justification and explanation to enhance clarity and understanding.

- Could the authors elaborate on why the Concentric Mixture of Mallows (CMM) and Concentric Mixture of Plackett-Luce (CMPL) models are suitable for capturing how different groups within a population rank a set of alternatives? Specifically, are there theoretical insights that support the choice of the CMPL model, given that CMPL outperform CMM in predicting complete rankings in the numerical experiments? Could there be scenarios where CMM performs better than CMPL, and if so, what characteristics of the data would lead to such differences?
- Have the authors experimented with fitting the models to datasets with more than three groups? If so, were there any noticeable patterns or performance changes with an increasing number of groups?
- In Figure 1, could the authors clarify why the dispersion parameters for the expert group are still widely distributed? Does this indicate variability in the expert group's preferences, or is it a limitation of the model?
- The conditions in Lemmas 3.3 and 3.4 are presented in a dense and abstract manner, making them difficult to interpret. Could the authors provide additional insights or intuitive explanations to help readers better understand these conditions and their implications?

**Questions:**

See above.

**Reviewer Confidence:**

2: The reviewer is willing to defend the evaluation, but it is likely that the reviewer did not understand parts of the paper

**Scope:**

3: The work is somewhat relevant to the Web and to the track, and is of narrow interest to a sub-community

---

### Official Review · Reviewer_RQhV · 2024-11-29

**Novelty:** 4
**Technical Quality:** 6

**Review:**

This paper concerns the problem of finding the ground truth ranking of $m$ objects from a collection of individuals who may only observe noisy or incorrect versions of the true ranking. The authors analyze an algorithm known as Surprisingly popular voting which can recover the ground truth ranking even when experts are in a minority. This algorithm is a variant of the surprisingly popular algorithm, which if applied naively to this problem leads to a large information requirements from each participant. However prior work found a succinct version for this purpose which is the main focus of this paper: the SP Voting algorithm.  In particular, they propose two noisy rank-order models and highlight conditions under which the new SP Voting algorithm can find the true underlying ranking. One of these models (Concentric Mixture of Mallows Models) has been proposed before but prior results only applied when the number of groups was limited to 2 (In this model there is a notion of how members in a group will rank compared to a true ranking). This paper extends to the case where there are more than 2 groups considered.



It also introduces a new model known as Concentric mixture of Plackett-Luce model. In this model, there is a parameter $\theta_g$ for each group and the probability of a particular ranking is given by the plackett-Luce model with parameter $\theta_g$. In the second model, they need the rankings to also stochastically dominate among the groups. In both of these rules, they show that the SP Voting algorithm has good sample complexity.
The algorithms for the Plackett-Luce Model seem to require an exponential sample complexity in $m$. The proofs are mostly differed to the appendix but seem correct. This seems quite large and unclear if the benefit of this algorithm is large compared to just eliciting a large amount of information from the surprisingly popular algoritm

They evaluate this on real world data-sets as well as generate new synthetic data-sets where the algorithm performs well.

The authors have done a reasonable job of looking at real-world as well as synthetic datasets. It seems reasonable that this will be of interest to the WWW communnity. However, I am not an expert in the area and am unsure how much technical novelty there is compared to earlier work such as [9]. Also the sample complexity bounds seem quite large so it is unclear if they are tight.

**Questions:**

N/A

**Reviewer Confidence:**

2: The reviewer is willing to defend the evaluation, but it is likely that the reviewer did not understand parts of the paper

**Scope:**

3: The work is somewhat relevant to the Web and to the track, and is of narrow interest to a sub-community

---

### Official Review · Reviewer_T7rd · 2024-12-01

**Novelty:** 4
**Technical Quality:** 4

**Review:**

**Overall Review**

This paper addresses the challenge of recovering ground truth rankings from individual reports in scenarios where experts are in the minority. Building upon the Surprisingly Popular (SP) algorithm, the authors extend the methodology to SP-voting and analyze its performance under two concentric rank-order models (Mallows and Plackett-Luce) with an arbitrary number of groups. The main contributions include the development of these new rank-order models, the identification of conditions for recovery of ground truth rankings, and empirical evaluations demonstrating the effectiveness of SP-voting compared to traditional methods.


**Paper Strength**

(1) **Innovative Methodology**: The paper introduces concentric mixtures of Mallows and Plackett-Luce models, which generalize existing models to accommodate more than two groups, allowing for a more nuanced understanding of voter behavior.

(2) **Theoretical Analysis**: The authors provide rigorous theoretical conditions under which SP-voting can recover ground truth rankings, including sample complexity analysis, which is crucial for understanding the practical applicability of their approach.

(3) **Empirical Validation**: The paper complements its theoretical findings with extensive empirical evaluations on both synthetic and real-world datasets, demonstrating that SP-voting outperforms traditional methods like the Copeland rule, especially in terms of sample complexity and accuracy.

**Paper Weakness**

(1) **Clarity of Exposition**: While the theoretical contributions are significant, some sections may be dense and complex, which could impede understanding for readers not deeply familiar with the underlying statistical models.

(2) **Limited Comparison with Baselines**: Although the paper compares SP-voting with the Copeland rule, a broader comparison with other widely-known ranking aggregation methods could strengthen the paper's claims about the superiority of SP-voting.

(3) **Lack of Detailed Implementation**: The paper could benefit from providing more detailed implementation guidelines or pseudo-code for the proposed models, which would aid in reproducibility.

**Questions:**

(1) Can you provide additional comparisons with other ranking aggregation methods beyond the Copeland rule to further substantiate the claims of SP-voting's effectiveness?

(2) Could you clarify the specific conditions under which the concentric rank-order models perform optimally, particularly regarding the number of groups?

(3) Are there any practical implications or recommendations for practitioners looking to implement SP-voting in real-world applications, based on your findings?

**Reviewer Confidence:**

2: The reviewer is willing to defend the evaluation, but it is likely that the reviewer did not understand parts of the paper

**Scope:**

4: The work is relevant to the Web and to the track, and is of broad interest to the community

---

### Official Review · Reviewer_W6QH · 2024-12-02

**Novelty:** 3
**Technical Quality:** 5

**Review:**

This paper studies the problem of recovering the truth under the SP voting scheme when the number of experts is small. The authors consider this problem under two models. Both theoretic and experimental results are provided.

The paper is well written and easy to follow in general. The studied problem is interesting. However, I find the proposed models too specific. It is not clear to me how well such models align with real-world applications. The paper also does not state clearly who has what information. For example, in section 2, if the prior P and the conditional probability Pr_s are both known to the algorithm, then reporting the individual rank alone seems sufficient, as the posterior of other voters can be inferred from this information. The other theoretic results align with my intuition once I accept the model, and the experiment results also look reasonable to me.

**Questions:**

What does the notation mean in line 334?

**Reviewer Confidence:**

3: The reviewer is confident but not certain that the evaluation is correct

**Scope:**

3: The work is somewhat relevant to the Web and to the track, and is of narrow interest to a sub-community

---

### Official Review · Reviewer_oo6U · 2024-12-03

**Novelty:** 5
**Technical Quality:** 5

**Review:**

The paper aims to address a fundamental problem, anticipated by Condorcet's theorem, how to use social choice (voting) to uncover the truth.

The paper looks at one approach, which aims to to distinguish between 'experts' and 'non-expects' by identifying a 'Surprisingly Popular' vote outcome, associating this outcome with the experts, and thus giving it higher weight.
The paper makes a rather deep analysis of the mathematical foundations of this approach, tries to improve it by dividing the voters into three classes (experts, intermediate, and non-experts) and explores the conditions under which this refined approach provides better results, as well as experimental data.

This is a hard-core, mathematically deep, social choice paper.   While the approach and analysis seems solid, evaluating it requires intimate familiarity with the background for this work and the specific approach taken, which I do not have.

Somehow, it seems to me that a hard-core social choice conference (or at least a conference with Social Choice as an official topic) would have easier time both finding reviewers with the necessary expertise to truly evaluate the paper, as well as an audience who can appreciate its novelty.

**Questions:**

Can you explain the relevance of the paper to this conference?

**Reviewer Confidence:**

2: The reviewer is willing to defend the evaluation, but it is likely that the reviewer did not understand parts of the paper

**Scope:**

3: The work is somewhat relevant to the Web and to the track, and is of narrow interest to a sub-community